# Quantification of Receptor Occupancy by Ligand—An Understudied Class of Potential Biomarkers

**DOI:** 10.3390/cancers12102956

**Published:** 2020-10-13

**Authors:** Suresh Veeramani, George J. Weiner

**Affiliations:** 1Holden Comprehensive Cancer Center, University of Iowa, Iowa City, IA 52241, USA; suresh-veeramani@uiowa.edu; 2Department of Internal Medicine, University of Iowa, Iowa City, IA 52241, USA

**Keywords:** RNA aptamers, LIRECAP, ligand–receptor complexes, cancer, biomarkers

## Abstract

**Simple Summary:**

Molecular complexes, such as those comprised of ligands such as hormones binding to their target receptors, are key determinants of health and disease. While research has focused on measuring receptors or ligands independently as biomarkers, very little attention has been given to measuring ligand-receptor complexes, in part, due to the limited availability of suitable technologies to do such measurements. This has led to underappreciation of ligand-receptor complexes as biomarkers in disease, including in cancer. In this commentary, the potential role of ligand-receptor complexes and their importance as biomarkers in cancer is discussed. We also describe a novel RNA aptamer-based technology, designated as ligand-receptor complex-binding aptamers (LIRECAP), that can provide precise measurement of the ligand occupancy of receptors and has potential use as a biomarker discovery platform.

**Abstract:**

Molecular complexes, such as ligand–receptor complexes, are vital for both health and disease and can be shed into the circulation in soluble form. Relatively little is known about the biology of soluble ligand–receptor complexes. The functional importance of such complexes and their potential use as clinical biomarkers in diagnosis and therapy remains underappreciated. Most traditional technologies used to study ligand–receptor complexes measure the individual levels of soluble ligands or receptors rather than the complexes themselves. The fraction of receptors occupied by ligand, and the potential clinical relevance of such information, has been largely overlooked. Here, we review the biological significance of soluble ligand–receptor complexes with a specific focus on their potential as biomarkers of cancer and other inflammatory diseases. In addition, we discuss a novel RNA aptamer-based technology, designated ligand–receptor complex-binding aptamers (LIRECAP), that can provide precise measurement of the fraction of a soluble receptor occupied by its ligand. The potential applicability of the LIRECAP technology as a biomarker discovery platform is also described.

## 1. Introduction

The central role of molecular complexes in biology has been well documented. Multimolecular complexes, including those formed by ligand–receptor interactions and multimeric receptor interactions, play a significant role in homeostasis; response to infection; and the pathogenesis of a variety of diseases, including cancer, by regulating cell growth and differentiation. Molecular complexes play a particularly important role in regulating immunity. Examples include complexes composed of cytokine–cytokine receptors, such as interleukins and their receptors; checkpoint molecules, including PD1-PDL1 complex; the CTLA-4-CD86 complex; and antigen-receptors, such as immune and TCR complexes [1,2,3,4,5,6,7,8]. Much therapeutic research is focused on altering ligand–receptor complexes, and on evaluating ligands or receptors individually as biomarkers. However, relatively little effort has been directed towards the measurement of the molecular complexes themselves for use as biomarkers of disease or for therapy. Traditional assays have limitations when being applied to quantify such complexes (Table 1).

Most such assays quantify complexes indirectly by measuring the levels of each molecular component independently or measuring the biological outcome of the complexes. Assays geared towards quantifying ligand–receptor complexes, such as FRET, are generally difficult to apply in a high-throughput platform like that required for analysis of a large numbers of samples or clinical diagnosis [1,17,18,19,20,21,22,23]. The result is that the biologically and clinically significant parameters associated with ligand–receptor complexes, such as the fractional occupancy of a receptor by its ligand, remain understudied and underappreciated. The ability to make such measurements directly—in the tissue, on cells or in the circulation—could provide unique information about the roles of receptors and ligands in biology, in diagnosis and in predicting or monitoring response to therapy.

In this review, we discuss the significance of molecular complexes, with specific focus on measuring the fractional occupancy of soluble receptor by ligand, in immune biology and in cancer. We also describe a novel RNA aptamer-based biomarker platform, designated for ligand–receptor complex-binding aptamers (LIRECAP), designed to measure fractional occupancy of receptor by ligand and the potential of this platform in cancer research and clinical management.

## 2. Soluble Ligand–Receptor Complexes

Much of our understanding of ligand–receptor biology comes from the studies on transmembrane receptors. As our knowledge of immune and cancer microenvironments has advanced, we have learned that many such receptors are shed from the cell surface into the tumor microenvironment and/or the circulation in soluble form. In general, cells that express high concentrations of a given receptor on their surface membrane are the source of soluble receptors. For example, soluble IL2Ra is shed mainly by CD4+IL2Ra+Foxp3+ Tregs and soluble IL6R is shed by activated CD4+IL6R+ effector T cells (Yoshida, Oda et al., 2013) [24]. The mechanism by which soluble receptors are generated is not fully understood. Proteinases appear to play an important role in cleaving surface membrane receptors. Several interleukin receptors, such as IL2Ra, IL6R and TNFRs, are cleaved off T cells by metalloproteinases, including ADAM17 (A Disintegrin And Metalloproteinase 17), neutrophil elastases and MMP9 (Matrix Metalloproteinase 9) [25,26]. Phagocytes, such as macrophages, may also play a role in this process [27]. Additional mechanisms that operate at the transcript level for soluble receptor production, including alternative transcript splicing, have also been reported [28].

Less is known about what role binding of ligand has on shedding of receptors or what happens to such receptors once they are shed. Unoccupied soluble receptors shed from the cells could bind to free ligands and circulate as ligand–receptor complex, as reported with IL1RA [29]. This requires that a high ligand concentration be present in the local microenvironment, which has been reported for several inflammatory conditions [30]. Alternatively, ligand–receptor complexes formed on the cell surface that are targeted for cleavage by proteinases as a regulatory mechanism may be shed as soluble complexes [31]. Regardless, presence of soluble receptors as ligand-bound complexes in the circulation and tissue microenvironment deserves greater attention, as it could play a key role in regulating a variety of cellular processes and could serve as a new class of biomarkers.

## 3. Signaling Mechanisms of Soluble Ligand–Receptor Complexes

Classically, when a ligand binds to the transmembrane receptor, adaptor proteins and intracellular kinases are recruited and a signaling process is initiated. Transmembrane receptors lacking signaling domains can interact with the nearby membrane-bound subunits with kinase activity in a cis manner to initiate signaling [15,26,32]. Soluble receptors lack the ability to mediate cis-signaling, as they are devoid of signaling domains and are not anchored to the cell membrane. On the other hand, soluble ligand–receptor complexes can initiate, augment or inhibit on-going signals. Much of our understanding of the effects of signaling mediated by ligand–receptor complexes comes from in vitro studies done using soluble cytokine receptors mixed with ligands. A summary of these mechanisms is shown in Figure 1.

## 4. Agonistic Activity

### 4.1. Initiation

Resting T cells do not express IL6R, but do express gp130, an IL6R subunit that is unable to bind IL6 directly. Thus, these resting cells do not bind to IL6 and are non-responsive to IL6 stimulation. However, soluble IL6-IL6R complexes in the microenvironment can interact with gp130 on resting cells and render these cells responsive to IL6 [33]. The existence of such trans-signaling by IL6-IL6R complex was confirmed in transgenic mice expressing both soluble human IL6R and human IL6. These double-positive mice display a massive hematopoiesis in extra-medullary sites when compared to human IL6+IL6R-mice or double-negative mice [34].

### 4.2. Sensitization

The IL2Ra receptor system includes a low-affinity receptor, namely, IL2Ra (Kd ≥ 10 nM), a medium-affinity receptor composed of IL2Rb and IL2Rg (Kd > 1 nM); and a high-affinity complex composed of IL2Ra, IL2Rb and IL2Rg (Kd < 0.1 nM). Resting T cells lack IL2Ra and are non-responsive to low levels of IL2 in the microenvironment [35]. However, IL2Ra-negative cells can respond to sub-nanomolar IL2 in the presence of soluble IL2Ra. Some cancer patients have high levels of soluble IL2Ra in their serum, with a significant fraction bound to IL2. It is therefore possible such soluble complexes in the circulation or tumor microenvironment of these cancer patients can induce generation of Tregs through trans-signaling from resting T cells [22,36].

## 5. Antagonistic Activity

Several studies have reported that soluble cytokine receptors, including soluble IL1RA, TNFR and IL12R, can inhibit signal transduction mediated by their respective ligands [10,12,13,32]. In these cases, soluble receptors inhibit binding of ligand to the surface receptor by competing for available cytokines or other interacting subunits, resulting in inhibition of the biological response [37]. Given the evidence that some soluble receptors, including IL1RA, can inhibit inflammatory disease conditions, such molecules are being considered as possible therapeutic agents [38,39,40].

Some soluble receptors, such as IL2Ra, can exhibit either agonistic or antagonistic properties depending on the circumstances. Soluble IL2Ra has been reported to both enhance IL2 signaling and compete for available IL2 and inhibit cell proliferation [22,41]. One potential explanation for this variable effect is that unoccupied soluble receptor and soluble ligand–receptor complexes mediate very distinct biological activity. Additional studies are required to understand these mechanisms more thoroughly. Nevertheless, these observations suggest understanding ligand occupancy of soluble receptors could be quite important.

## 6. Quantification of Ligand–Receptor Complexes and Fractional Occupancy

Monoclonal antibody (mAb)-based assays have long been the gold standard for diagnostic assays assessing receptor or ligand levels. For many mAbs, it is not clear how the presence of ligand impacts binding of that mAb to its target receptor. In addition, Ab-based assays are not designed to determine the fraction of receptors or ligands that are bound to each other. Classic generation of mAbs rarely leads to identification of epitopes formed due to conformational changes resulting from the interaction between receptors and ligands [42]. This is in large part because such changes get denatured during processing by professional antigen-presenting cells [43,44]. Alternative approaches, including phage display and other approaches to generating synthetic antibodies, are being explored to counter this problem, but their diagnostic applicability to detecting ligand–receptor complexes remains unclear [44].

## 7. A Novel Assay Platform Based on LIRECAP

We have been exploring the use of RNA aptamers to measure molecular complexes in biospecimens as a way of overcoming many of these challenges. This assay has been designated as the LIRECAP assay [36]. RNA aptamers that are used as the basis for the LIRECAP assay have a number of advantages over mAb as tools to assess such complexes. Selection of aptamers is not dependent on antigen processing, and can include selection based on binding to epitopes in their natural conformation. Even subtle changes in conformation and protein charge can be targeted with aptamers [45,46]. Aptamers whose binding is dependent on the interaction between ligand and receptor, as used in LIRECAP assay, can be selected using either whole cells or purified ligand–receptor complexes. The SELEX approach and its modifications allow for a negative selection to remove unwanted aptamers before a positive selection is used to enrich the desired aptamers. Use of sequencing and bioinformatics allows for identification of large numbers of potentially valuable sequences that can then be synthesized and tested at a fraction of the cost and time necessary to generate and test mAbs.

The LIRECAP assay is based on pairs of RNA aptamers, where one aptamer binds preferentially to a receptor in its unoccupied form and the second aptamer binds preferentially to the ligand–receptor complex [36]. The overall scheme of the LIRECP assay is summarized in Figure 2. The key steps in this technology are as follows:Aptamer selection: A pair of RNA aptamers are identified, where one aptamer binds preferentially to the ligand–receptor complex and the second aptamer binds preferentially to the unoccupied receptor. Differences in their variable regions are responsible for this distinct binding. For the LIRECAP assay to be effective, such aptamers should be of the same length and have the same 5′ and 3′ primer-binding regions. The selected aptamers should not cross block each other, nor interfere with the binding of the ligand to the receptor.Aptamer addition to biospecimens: The pair of aptamers identified above is added in equimolar concentrations to a biospecimen and unbound aptamers are removed by washing.Aptamer quantification: Bound aptamers are extracted using standard molecular biology procedures and quantified using a standard TaqMan RT-qPCR reaction. The TaqMan amplification utilizes a set of PCR primers that bind to the 5′ and 3′ ends of both aptamers, thus amplifying both aptamers proportionately. TaqMan probes that bind to the individual variable central regions of the aptamers, each labeled with a different color, allow for quantification of each of the aptamers individually and for calculation of their ratio of binding to the samples.Use of a standard curve to calculate fractional occupancy: A standard curve is constructed based on the fractional occupancy of receptor by ligand using control samples with known fractional occupancy. Results of the test samples are compared to the standard curve. Using the binding ratio provides an internal control so the specificity of the two aptamers (for ligand–receptor complex and unoccupied receptor, respectively) does not have to be absolute, only relative. Thus, by using two aptamers and a standard curve, the relative binding of the two aptamers precisely reflects the fractional occupancy of the receptor by ligand.

There are a number of logistical advantages to the LIRECAP assay. It can be performed using equipment that is standard in both research and clinical laboratories. Once selected, aptamers and probes can be produced inexpensively. The nucleic acid nature of aptamers allows for high-throughput handling of samples using standard molecular biology techniques.

We first applied the LIRECAP assay to measuring the IL2-IL2Ra ligand–receptor complex [36]. Aptamers were generated using a whole cell-based SELEX system with CD4+IL2Ra+ Treg cells from normal human donors. Each round of selection was done using cells from a different donor. Aptamers that increased in prevalence after each round of selection were selected for further evaluation. Five of the most prevalent aptamers were specific to IL2Ra. While the Tregs used in the selection process were not treated with IL2, it is likely that some of the surface IL2Ra molecules on the Tregs were occupied by IL2 and other IL2Ra molecules were unoccupied. This explains why some of the identified aptamers preferentially bound to the IL2-IL2Ra complex while others preferentially bound to unoccupied IL2Ra. More specifically, the aptamers designated Tr-1 and Tr-7 had greater affinity and binding towards the IL2-IL2Ra complex, compared to unoccupied IL2Ra, while the aptamers designated Tr-6 and Tr-8 had better affinity to and bound preferentially to unoccupied IL2Ra. These aptamers did not cross block each other or block binding of IL2 to IL2Ra. Thus, they fit the criteria outlined for the LIRECAP assay. Further evaluation using Tr-7 and Tr-8 demonstrated that the assay performed as designed. The standard curve generated using known factional occupation of IL2Ra by IL2 was linear and was successfully used to calculate fractional occupancy in unknown samples assayed in parallel.

## 8. Measuring Soluble IL2-IL2Ra Complexes in Cancer Biospecimens Using the LIRECAP Assay

Initial studies exploring fractional occupancy of IL2Ra by IL2 in the serum of healthy individuals and cancer patients, including those with lymphoma and melanoma, demonstrated the LIRECAP assay can be used to calculate fractional occupancy of IL2Ra by IL2 across a spectrum of receptor concentrations. As summarized in Figure 3 and in [36], there was wide variability in fractional occupancy of IL2Ra by IL2 among individuals, including both healthy individuals and cancer patients, ranging from 27% to 85%. Patients with lymphoma and metastatic melanoma in general had lower fractional occupancy than the healthy individuals. Serum from lymphoma patients with Diffuse Large B Cell Lymphoma phenotype had lower fractional occupancy than those with Follicular Lymphoma phenotype. These preliminary data speak to the technical practicality and the potential value of the LIRECAP assay. These data also speak to the potential of the LIRECAP assay to identify unexpected biology and clinically useful biomarkers. Additional studies with larger numbers of lymphoma biospecimens, and specimens from patients with a variety of other cancers and inflammatory disorders, are on-going.

## 9. Potential of the LIRECAP Assay with Other Ligand Receptor Pairs

The LIRECAP assay done using the IL2-IL2Ra system supports its development to assess ligand occupancy of receptor in other ligand–receptor systems. Examples include cytokine and cytokine receptors such as IL6 and IL6R, and immune checkpoints such as PD1 and PDL1, where receptors and/or ligands alone are showing promise as biomarkers in both malignancy and inflammatory conditions [32,47,48,49]. No information is available on the levels of ligand occupancy of soluble receptors in these systems.

## 10. Future Directions and Conclusions

Lord Kelvin is reported to have said “If you cannot measure it, you cannot improve it”. This is applicable to our understanding of the role ligand–receptor complexes play in biology in general and cancer in particular. This field has received little attention, because there has been no easy way to quantify such complexes. We are hopeful the LIRECAP assay will help open this potentially rich field of discovery to further research. The target options for the LIRECAP assay are nearly endless. The value of this assay platform in each case will be dependent on the scientific or clinical importance of the ligand–receptor complex, the stability of the complex in clinical biospecimens and the ability to select aptamers for the analysis. Development of aptamers against various targets and their use in a broad range of scenarios will be a required before we know the true potential of the LIRECAP assay as a broader diagnostic platform. Our studies to date with the LIRECAP assay have focused on soluble ligand–receptor complexes. A similar aptamer-based technology could be applicable to analysis of receptor occupancy on the surface of cells or in tissue, where we know little about how fractional occupancy impacts on biology or response to therapy. Furthermore, it could be applied to other biological systems, such as receptor multimerization, where quantifying multimolecular interaction would be useful.

## Figures and Tables

**Figure 1 cancers-12-02956-f001:**
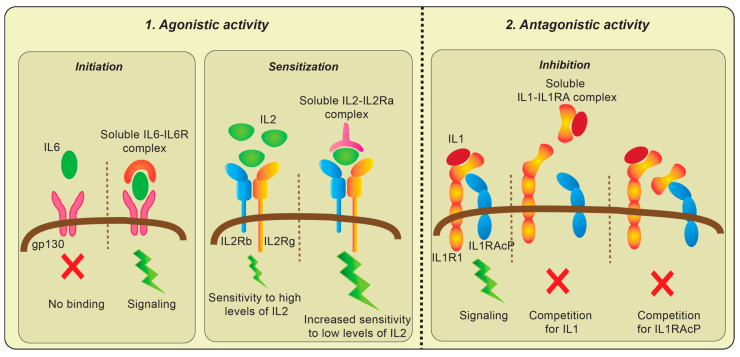
Mechanisms of action by soluble receptors and ligand–receptor complexes. Schematic representation of the mechanisms by which soluble receptors and ligand–receptor complexes influence membrane-bound receptor components and regulate signal transduction and cellular activity.

**Figure 2 cancers-12-02956-f002:**
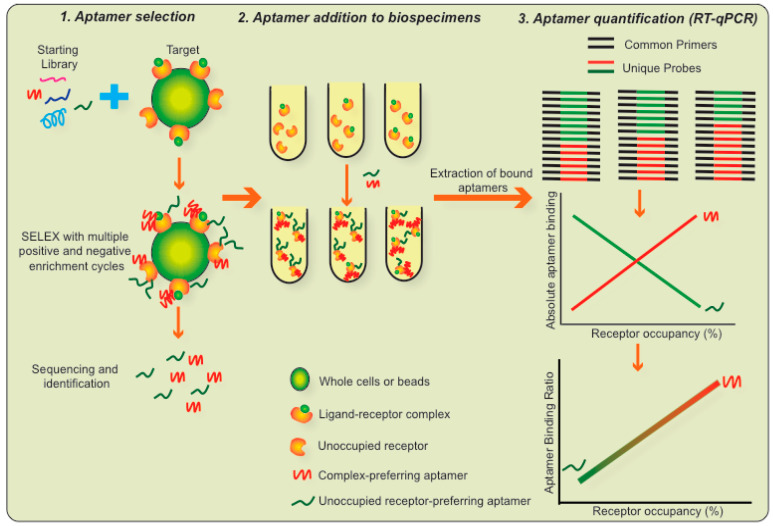
Key steps in the ligand–receptor complex-binding aptamers (LIRECAP) technology for measuring the fractional occupancy of soluble receptors. LIRECAP technology is based on the aptamer pairs that preferentially bind to either the unoccupied receptor or the ligand–receptor complex. When added in equimolar concentration to a given sample, the ratio of binding linearly correlates with the fraction of receptor occupied by the ligand in that sample. Using a standard curve made with samples of known fractional occupancy values, it is possible to calculate the fractional occupancy of the receptor in an unknown sample, such as clinical biospecimens.

**Figure 3 cancers-12-02956-f003:**
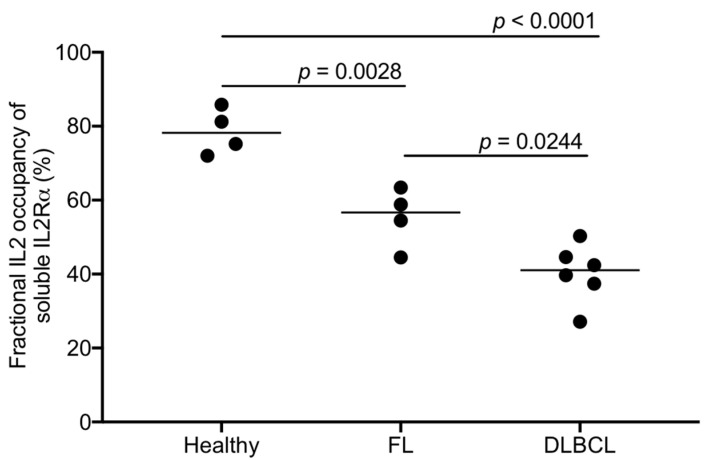
Fractional IL2 occupancy of soluble IL2Ra in healthy individuals and lymphoma patients, as analyzed by LIRECAP assay. Fractional occupancy of soluble IL2Ra by IL2 was measured using the LIRECAP assay using serum from healthy subjects and patients with Follicular lymphoma (FL) or Diffuse large B cell lymphoma (DLBCL). Data was compared using one-way ANOVA (Tukey’s test).

**Table 1 cancers-12-02956-t001:** Current technologies and their applications in ligand–receptor studies.

Primary Application	Technique(s)	Limitations	Citations
Quantify receptors or ligands	RIA and ELISAs using labeled antibodies	Usually used to measure individual receptors or ligands; ELISA-based assays that combine both anti-receptor and anti-ligand antibodies to detect complexes are designed to quantify complexes and not assess fractional occupancy of a receptor by a ligand	[9]
Determine the ligand–receptor interaction kinetics	Labeled ligands (e.g., radioactively labeled), Surface Plasmon Resonance (SPR) and Fluorescence Resonance Energy Transfer (FRET)	Limited use in clinical diagnosis due to challenges in handling large number of samples simultaneously	[10,11]
Identify interacting molecules	Co-immunoprecipitation; mass spectrometry	Data is qualitative to semi-quantitative; high-throughput assay is not feasible	[12,13]
Determine co-localization in the cellular environment	Confocal microscopy	Data is qualitative	[14]
Measure biological function	Assessment of signal transduction (e.g., arrays, such as ZeptoMARK), cell proliferation or differentiation induced as a result of ligand–receptor binding	Provides indirect analysis of ligand–receptor interaction	[2,15,16]

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
