# Peer review of "Quantification of Receptor Occupancy by Ligand—An Understudied Class of Potential Biomarkers"

_cancers, 2020, doi:10.3390/cancers12102956_

Round 1

Reviewer 1 Report

All my comments have been addressed

Reviewer 2 Report

More revisions were requested, but I can accept this version without further comments.

Reviewer 3 Report

I read the comments the authors made in response to my suggestions and
they were just perfect, including the reproducibility of the assay. I
still hope the paper is published.

This manuscript is a resubmission of an earlier submission. The following is a list of the peer review reports and author responses from that submission.

Round 1

Reviewer 1 Report

A perspective 'Quantification Of Receptor Occupancy By Ligand – An Understudied Class of Potential Biomarkers' by Veeramani and Weiner is an interesting short review about aptamers to ligand-receptor complexes. The overall estimation of the review is high. However, there are several concerns to be improved before the publication.

1) Contrary to statements in the introduction, the content of ligand-receptor complexes can be estimated with ELISA or some other antibody-based technique with two types of antibodies. For example, see ref [9].

2) Page 5: 'Relative to mAbs, even subtle changes
140 in conformation and protein charge can be targeted with aptamers'. This statement is inappropriate. Many aptamers bind several related proteins. And some antibodies are highly sensitive to protein conformation.

3) The sole example of LIRECAP success (ref.36) includes minor differences in Kd between free protein and its complex. Authors used pairs of aptamers to provide reliable results. I think, these difficulties are to be discussed, and minimal requirements for aptamer selectivity are to be proposed.

Author Response

Thanks

Suresh Veeramani and George J. Veeramani

Reviewer 2 Report

This review article is well written, and there is no doubt about the clinical importance of accurate detections of ligand-bound complexes for diagnostic and therapy predictions of patients suffering of malignancies and/or inflammatory diseases.

The described cytokines and immune checkpoint molecules are of extreme importance in view of personalized immune therapies nowadays in clinical use and becoming more and more specific and developed for more indications. 

Table 1 represents a schematic representation of known techniques used to analyze ligand-receptor expression levels. Here I miss multifactorial quantitative detection techniques as the one of ZeptoMARK (https://lbb.ethz.ch/equipment/equipment/zeptoreader.html).

Even though the article explains well the role of soluble ligand-receptor complexes, the main message addresses the previously published LIRECAP technique already published several times by the same two authors.

Moreover there are sentences such as the one on line 75-78 “Regardless, presence of soluble receptors as ligand-bound complexes in the circulation and tissue microenvironment deserves greater attention as they could play a key role in regulating a variety of cellular processes signal transduction and could serve as a new class of biomarkers.», which refer to the tissue microenvironment. There is no evidence that the LIRECAP technique developed by Veeramani and Weiner works on tissue and this should be implemented or the text should be adapted.

Author Response

Thanks

Suresh Veeramani and George J. Weiner

Reviewer 3 Report

Dear authors,

I have worked with aptamers from the beginning. Thirty years. I expected to not see much of interest and I was wrong - your paper has much to recommend it. In a sense your paper is an attempt to generalize your idea that "receptor occupancy" is a strong potential biomarker for many conditions, and that we ought to use quantification of receptor occupancy as frequently as possible. I agree with you.

The problem you face (as do all of us who work on any piece of quantitative proteomics) is one of scale! You worked very hard to sort out the receptor occupancy for one receptor and one ligand. Probably human biology uses about 4,000 receptors and a larger number of ligands. In addition many receptors function as dimers, and often as heterodimers. You have proposed that we solve a large combinatoric problem. I would like you to add a paragraph toward the end of the manuscript in which you propose how to scale what you have done so that you need not know what receptor/ligand pair is important but rather let the data emerge with that answer.

As you know many human proteins have had aptamers made against them - probably the number is more than 7,000 at this moment. Perhaps one way to get to scale will be to use existing aptamers as one of the pair you use in your assay. I leave that up to you.

I have one semi-serious desire for more data, either for this paper or for a subsequent paper (your choice). I would like to know if for your Figure 3 what the repeats of the same sample would look like. I have had a lot of experience with proteomics using aptamers and the CVs really matter. It would be terrific to see what would happen if you did one sample 50 times so we knew how much variance you would see.

That is a quibble. I liked the paper very much.

Author Response

(The authors gave the same response as above.)
